

# Temporal expression profiles of lncRNA and mRNA in human embryonic stem cell-derived motor neurons during differentiation

Xue-Jiao Sun, Ming-Xing Li, Chen-Zi Gong, Jing Chen, Mohammad Nasb, Sayed Zulfiqar Ali Shah, Muhammad Rehan, Ya-Jie Li and Hong Chen

Department of Rehabilitation Medicine, Tongji Hospital, Tongji Medical College, Huazhong University of Science and Technology, Wuhan, China

## ABSTRACT

**Background**. Human embryonic stem cells (hESC) have been an invaluable research tool to study motor neuron development and disorders. However, transcriptional regulation of multiple temporal stages from ESCs to spinal motor neurons (MNs) has not yet been fully elucidated. Thus, the goals of this study were to profile the time-course expression patterns of lncRNAs during MN differentiation of ESCs and to clarify the potential mechanisms of the lncRNAs that are related to MN differentiation.

**Methods**. We utilized our previous protocol which can harvest motor neuron in more than 90% purity from hESCs. Then, differentially expressed lncRNAs (DElncRNAs) and mRNAs (DEmRNAs) during MN differentiation were identified through RNA sequencing. Bioinformatic analyses were performed to assess potential biological functions of genes. We also performed qRT-PCR to validate the DElncRNAs and DEmRNAs.

**Results**. A total of 441 lncRNAs and 1,068 mRNAs at day 6, 443 and 1,175 at day 12, and 338 lncRNAs and 68 mRNAs at day 18 were differentially expressed compared with day 0. Bioinformatic analyses identified that several key regulatory genes including POU5F1, TDGF1, SOX17, LEFTY2 and ZSCAN10, which involved in the regulation of embryonic development. We also predicted 283 target genes of DElncRNAs, in which 6 mRNAs were differentially expressed. Significant fold changes in lncRNAs (NCAM1-AS) and mRNAs (HOXA3) were confirmed by qRT-PCR. Then, through predicted overlapped miRNA verification, we constructed a lncRNA NCAM1-AS-miRNA-HOXA3 network.

## INTRODUCTION

Generation of specific cell types from human embryonic stem cells (hESC) in vitro have provided powerful platforms to study human disease and to understand fundamental biological processes. Highly efficient directed differentiation of hESCs into spinal motor neurons have been used to explore not only spinal motor neuron (MN) development but also MN disorder mechanisms (*Chen et al., 2014*; *Du et al., 2015*). Motor neurons are

Corresponding author
Hong Chen, chen-hong1129@hotmail.com

responsible for innervating skeletal muscles in the periphery and controlling movement. Transcription factors (TFs) regulate precise temporal and spatial gene expression in motor neuron specification and differentiation (*Cave & Sockanathan, 2018*; *Alaynick, Jessell & Pfaff, 2011*). The TFs Olig2 and Ngn2 function in opposition to regulate gene expression in MN progenitors in the pMN domain and the TFs Isl1 and Lhx3 are crucial for specifying MN identity (*Thaler et al., 2002*; *Seo, Lee & Lee, 2015*; *Lee et al., 2005*).

LncRNAs, ranging in length from 200 nt to 100 kb, are highly expressed in the central nervous system. Accumulating evidence suggested that lncRNA played crucial roles in numerous biological and pathological processes at the chromatin remodeling level, transcriptional level and post-transcriptional level (*Mercer, Dinger & Mattick, 2009*; *Kopp & Mendell, 2018*). Notably, lncRNAs function as key regulators of cell differentiation and development, especially in neurogenesis. In particular, lncRNAs can regulate ESC pluripotency and control multiple lineage differentiation by association with miRNAs, RNA-binding proteins, and epigenetic modifiers (*Loewer et al., 2010*). LncRNA-1604 functioned as competing endogenous RNAs (ceRNAs) of miR-200c and indirectly regulated the core TFs ZEB1 and ZEB2 during neural differentiation from mouse ESCs (*Weng et al., 2018*). LncRNA Haunt functions as a genetic enhancer and an epigenetic repressor of HOXA gene activation during ESC differentiation (*Yin et al., 2015*). *Dlk1-Dio3* locus-derived lncRNAs play a critical role in maintaining postmitotic MN cell fate by repressing progenitor genes and they shape MN subtype identity by regulating Hox genes (*Yen et al., 2018*). Nevertheless, at present very little functional characterization of lncRNAs in human motor neuron differentiation has been elucidated.

We used spinal motor neuron differentiation to profile the temporal changes without further purification steps. We combined highly efficient MN differentiation of hESCs in vitro with RNA-seq analysis to reveal the expression profiles of lncRNAs and mRNAs. Our findings may provide a new theoretical basis for further studies on lncRNAs modulation of motor neuron differentiation.

## MATERIALS AND METHODS

### Cell cultures from embryonic stems cells to spinal motor neurons

H9 (WA09, NIH registry 0046) hESC lines were obtained from WiCell Research Institute (Madison, WI). Human ESCs were maintained on irradiated mouse embryonic fibroblasts and differentiated as described before (*Du et al., 2015*). Briefly, ESCs were cultured using MN differentiation medium containing DMEM/F12, Neurobasal medium, N2, B27, ascorbic acid, Glutamax and penicillin/streptomycin (All from Gibco). The medium was additionally supplemented with chemical compounds: 3 µM CHIR99021 (Torcris), 2 µM DMH-1 (Torcris) and 2 µM SB431542 (Stemgent) for 6 days differentiated into neuroepithelial progenitors; 1 µM CHIR99021, 2 µM DMH-1, and 2 µM SB431542, 0.1 µM RA (Sigma) and 0.5 µM Purmorphamine (Pur, Sigma) for 6 days differentiated into MN progenitors; 0.5 µM RA and 0.1 µM Purmorphamine for 6 days differentiated into MNs.

**Table 1  Primer sequences of five lncRNAs and mRNAs.**

| Gene | Forward | Reverse |
|------|---------|---------|
| GAPDH | GGAAGCTTGTCATCAATGGAAATC | TGATGACCCTTTTGGCTCCC |
| LncRNA rp11.001 | CAGCCCAAGGAACATCTCACC | TCTTGCCAACTTGAGTGTCCAT |
| LncRNA rp11.003 | ATCGGACTGTTCAACTCACCTG | TCAGCCGCTAAGCCAAGAAG |
| NCAM1-AS | TGAGATGCGAGACCTCCAGAC | CTCCAACTGCCTCATTATCCG |
| LncRNA rp11-c9-001 | GGGGGCTGGAAACCAACTTAT | CATCCCAAGTCCAGCGTGAA |
| H19 | CGGCCTTCCTGAACACCTTA | GTGTCTTTGATGTTGGGCTGATG |
| ZSCAN10 | GCCACCGTTTCCGCAATA | GCAGGTGTCGCAGCAGATT |
| CST1 | CCCCAAGGAGGAGGATAGGAT | AGTTGGGCTGGGACTTGGTA |
| VRTN | TCCCGCTCAACCTACTATGCC | CGTTTGAAGCAGCGATAGGG |
| OCT4 | TCTATTTGGGAAGGTATTCAGCC | CCTCTCACTCGGTTCTCGATACTG |
| HOXA3 | CTCAGAATGCCAGCAACAACC | ACAGGTAGCGGTTGAAGTGGA |
| SP9 | CCAAGCAGTTTTTCCGAGCAG | GGCTCGTGTTGCCGATCTT |

## Total RNA isolation and RNA sequencing

Total RNA from samples was extracted using the Trizol reagent (Invitrogen). RNA quantity and purity of total were confirmed using NanoDrop ND-2000 spectrophotometer (Thermo Fisher Scientific, USA). RNA integrity was assessed by Agilent 2100 Bioanalyzer (Agilent Technologies, USA). Ribosomal RNA was removed. Sequencing libraries were constructed. The dTTP were replaced by dUTP in the reaction buffer during second strand cDNA synthesis. Products were purified and library quality was assessed on the Agilent Bioanalyzer 2100 system. RNA Sequencing was performed on the Illumina HiSeq ((Illumina, USA) by KangChen Biotechnology Corporation (Shanghai, China) using next-generation sequencing analysis. Raw data of the RNA sequencing have been deposited in the Gene Expression Omnibus (GEO) public database (accession number: GSE151744).

## Immunocytochemistry

The primary antibodies information was used: SOX1 (gIgG, 1:1000, R&D), HOXA3 (mIgG, 1:1000, R&D), OLIG2 (rIgG, 1:1000, Millipore), HB9 (rIgG, 1:1000, Millipore), NF-200 (mIgG, 1:400, CST). Image J was used to perform cell counting (NIH, USA).

## Quantitative real-time PCR

Total RNA was extracted using the Trizol reagent (Invitrogen). qRT-PCR was then performed and the $2^{-\Delta\Delta Ct}$ method was calculated for quantification. The GAPDH was used as an internal control. The primer sequences used are listed in Table 1.

## Transfection of lncRNA Smart Silencer

Motor neuron progenitors were transfected with lncRNA Smart Silencer (RiboBio Co., Guangzhou, China) to knock down the expression of NCM1-AS. The lncRNA Smart Silencer contained a mixture of three siRNAs and three antisense oligonucleotides. The target siRNA sequences were as followed: 5′-ACAACCCGATGACAGCAGA-3′, 5′-CCAAATGGAGAACGTGCAA-3′and 5′-GTACTCGGTCTTTGCTGGC-3′. The target

antisense oligonucleotides sequence were as followed: 5′-ATGAAAGGAAAGGCACCAGC-3′, 5′-ACATCTAACAAGGAGGACAC-3′, 5′-AGGTTGACCGCAATGCACAT-3′. The negative control (NC) Smart Silencer did not contain domains sequences homologous to those of humans, rats, or mice. The cells were transfected with the lncRNA Smart silencer using Lipofectamine 3000 (Invitrogen, USA) and collected 48 h after transfection for RNA isolation.

### Bioinformatic analysis

Gene Ontology (GO) analysis was used to investigate differentially expressed mRNAs with GO categories. The predicted target genes above were conducted using the DAVID database (http://david.abcc.ncifcrf.gov/). GO terms with a $P$ value <0.05 were considered as significantly enriched. PPI networks was used STRING database (https://string-db.org/) and Cytoscape. The networks were visualized in CytoHubb plug-in of Cytoscape. LncRNAs and mRNAs possessing microRNA recognition elements (MREs) for the targeted miRNAs were predicted using the miRanda and TargetScan.

### Statistical analysis

All qRT-PCR results are expressed as the means $\pm$ SEM of at least three independent experiments. Statistical analyses were performed with SPSS statistics software version 22.0. $P$ values < 0.05 was considered statistically significant. All graphs were made with GraphPad Prism 8.

## RESULTS

### Differentiation of high purity motor neuron from human embryonic stem cells

In this study, we used hESCs to differentiate into spinal cord MNs *in vitro*. The MNs were generated using chemical protocol described as method part (*Du et al., 2015*) (Fig. 1A). The hESCs can differentiate into SOX1 neuroepithelial progenitors at day 6, OLIG2 motor neuron progenitors at day 12, and HB9 motor neuron at day 18 high efficiently by using a combination of small molecules (Figs. 1B–1D).

### Patterns of gene expression changes from hESCs to motor neuron

High throughput sequencing is an efficient approach for investigating the biological function of RNAs. All the differently expressed DElncRNAs and DElncRNAs were statistically significant ($P < 0.05$) with fold change (FC) > 2.

A total of 441 DElncRNAs (192 up-regulated and 249 down-regulated), 443 DElncRNAs (198 up-regulated and 245 down-regulated) and 338 DElncRNAs (164 up-regulated and 174 down-regulated) were identified in D6 *vs* D0 (neuroepithelial progenitors, NEP), D12 *vs* D0 (motor neuron progenitors, MNP), and D18 *vs* D0 (motor neuron, MN) respectively (File S1). The volcano plots of expression profile for all the detected transcripts showed the relationship between the fold change and the significance (Figs. 2A–2C). To determine the key RNAs in human MN differentiation, we analyzed DElncRNAs in Venn diagram form. Veen analysis revealed that 33 lncRNAs were simultaneously up regulated and 78 were down regulated on D6, D12 and D18 (Figs. 2D–2E). Meanwhile, a total of 1068

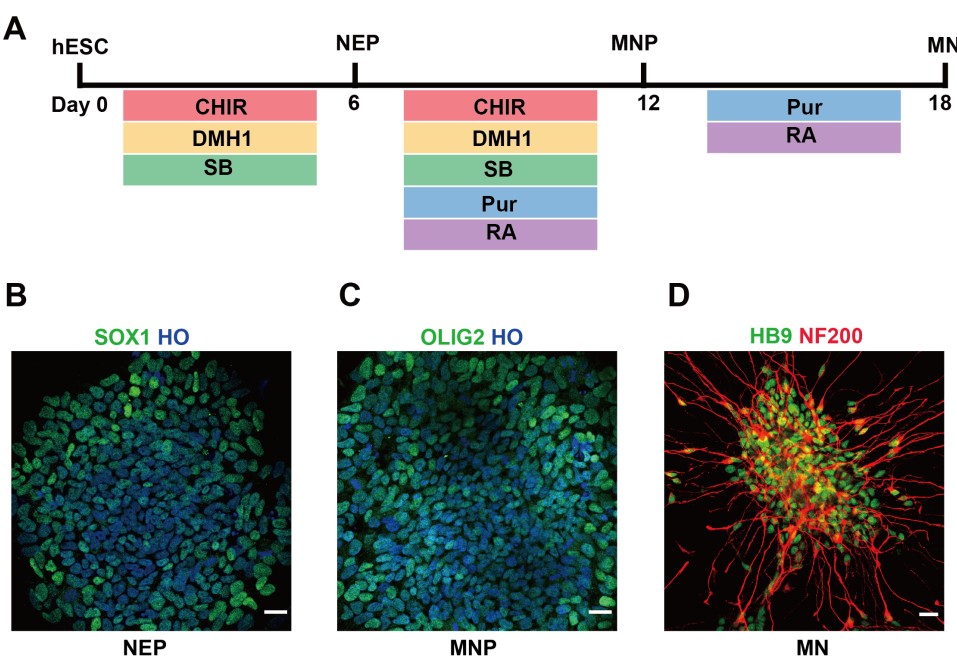

**Figure 1 Differentiation of hESCs into MNs.** (A) Schematic diagram of the protocol used to obtain MNs from hESCs. hESCs, human embryonic stem cells; NEPs, neuroepithelial progenitors; MNPs, motor neuron progenitors; MNs, motor neurons. (B–D) hESCs differentiated to SOX1+ NEPs at day 6, OLIG2 + MNPs at day 12 and HB9+ MNs at day 18, respectively. Scale bar = 50 μm.

DEmRNAs (360 up-regulated and 708 down-regulated), 1175 mRNAs (444 up-regulated and 731 down-regulated) and 68 DEmRNAs (34 up-regulated and 34 down-regulated) were identified in D6 *vs* D0, D12 *vs* D0 and D18 *vs* D0 respectively (File S2). The volcano plots was shown in Figs. 3A–3C. Veen analysis revealed that 8 mRNAs were simultaneously up-regulated and 34 down-regulated (Figs. 3D–3E).

## Validation of lncRNAs and mRNAs expression

To verify the results of the RNA sequencing, five strongly expressed DElncRNAs and DEmRNAs during MN differentiation with FC > 40 were selected for qRT-PCR validation. As shown in Figs. 4A–4E, the expression of ZSCAN10, OCT4 and VRTN were down regulated, while HOXA3 and SP9 were up regulated. Moreover, the expression of ENST00000454596 (lncRNA rp11.001), ENST00000419695 (lncRNA rp11.003) and ENST00000583521 (lncRNA rp11-c9-001) were down regulated, while NCAM1-AS were up regulated (Figs. 4F–4J). However, lncRNA H19 increased at D12, but decreased significantly at D18. Our results agreed with the data of RNA-sequencing generally.

## Bioinformatics analysis during MN differentiation

To identify the key factors that regulated MN differentiation, GO analysis of DEmRNAs was performed on three different aspects namely biological process (BP), molecular function (MF) and cellular component (CC) shown in Fig. 5A. In the BP domain, the top 6 GO terms were enriched in cell migration involved in gastrulation, somatic stem

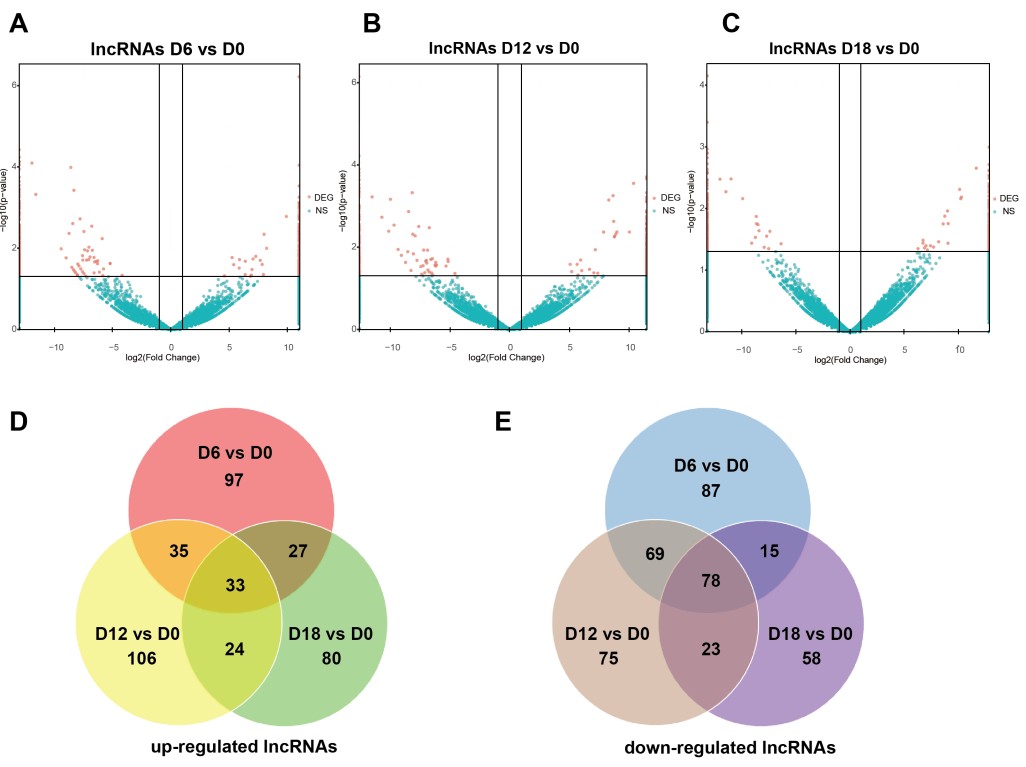

**Figure 2** **Expression Profiles of lncRNAs.** (A–C) Plots indicate up-regulated and down-regulated lncR-NAs at different stages points of MN differentiation from ESCs. (D–E) Venn diagrams show the number of overlap lncRNAs during the different stages of MN differentiation from ESCs.

cell population maintenance, positive regulation of transcription from RNA polymerase II promoter, positive regulation of cell proliferation, cardiac cell fate determination and anterior/posterior pattern specification. Genes associated with cell migration involved in gastrulation were SOX17, MIXL1 and CER1. Genes associated with positive regulation of cell proliferation were EPHA1, ETS1, HOXA3, POU3F3, FLT1, and TDGF1. In the CC domain, the top 3 GO terms were transcription factor complex, membrane raft and nucleoplasm. In the MF domain, the top 3 GO terms were sequence-specific DNA binding, transcription factor activity and HMG box domain binding. Figure 5B showed the heat map of DEmRNAs.

Furthermore, the Protein-Protein Interaction (PPI) network of DEmRNAs contained 28 nodes and 39 edges (Fig. 5C). The topological analysis of the network was carried out by Network Analyzer in Cytoscape. PPI network was imported into Cytohubba to determine the hub transcription factors with high degree of connectivity between the nodes. The top ten hub genes were POU5F1, TDGF1, SOX17, LEFTY2, ZSCAN10, CER1, ZFP42, MIXL1, L1TD1 and ESRP1 shown in Fig. 5C.

## Analysis of lncRNAs target mRNAs

LncRNA may regulate nearby protein-coding genes by cis-regulatory effects. We analyzed potential function of the target genes of DElncRNAs at D18 *vs* D0. Total 338 DElncRNAs
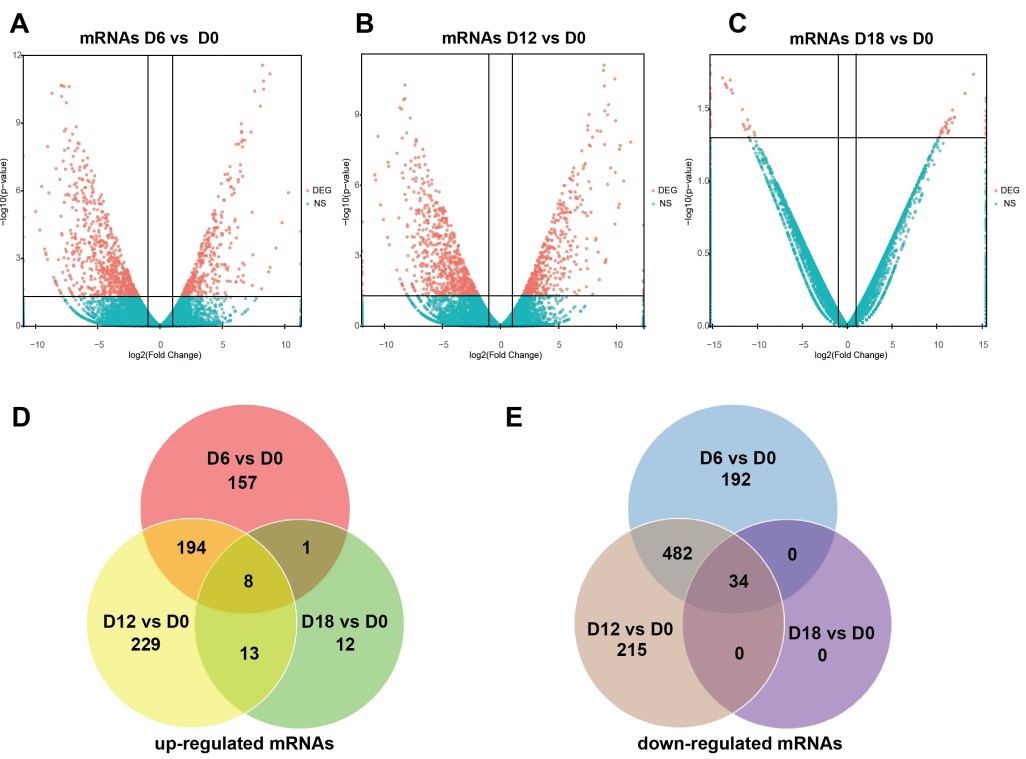

**Figure 3** **Expression Profiles of mRNAs.** (A–C) Plots indicate up-regulated and down-regulated mR-NAs at different stages points of MN differentiation from ESCs. (D–E) Venn diagrams show the number of overlap mRNAs during the different stages of MN differentiation from ESCs.

had 289 mRNAs target genes (File S3). Then we performed GO analysis on DElncRNAs target genes to explore their potential biological functions. The GO terms related to BP, CC and MF were shown in Fig. 6A. The most enriched BP term was related to positive regulation of telomerase activity. As for CC, we found that the most enriched term were nucleoplasm and transcription factor complex. As for MF, transcription factor activity, sequence-specific DNA binding and metal ion binding were most enriched. Moreover, KEGG analysis was made in Fig. 6B and the most enriched 5 top pathway terms were MAPK signaling pathway, Ribosome, Pyrimidine metabolism, Hippo signaling pathway and Cushing syndrome. Veen diagram analysis indicated that 6 mRNAs targeted by DElncRNAs were found at the interaction of DEmRNAs at D18 *vs* D0 (Fig. 6C). Moreover, lncRNAs and their *cis* target DEmRNAs were up-regulated at D18, including two well-known TFs HOXA6 and HOXC9. We also performed the topological analysis of PPI network on these target genes (Fig. 7), indicating the highly connected hub nodes in PPI network.

## Construction of lncRNA-miRNA-mRNA interaction network

LncRNAs can regulate gene expression by acting as ceRNAs to sponge miRNAs (*Thomson & Dinger, 2016*). Thererfore, we constructed a ceRNA interaction network from verified DE mRNAs and DE lncRNAs based on previous qRT-PCR data. As the RNA sequencing and PCR data shown, the transcription factors HOXA3 and SP9, exhibited continuous

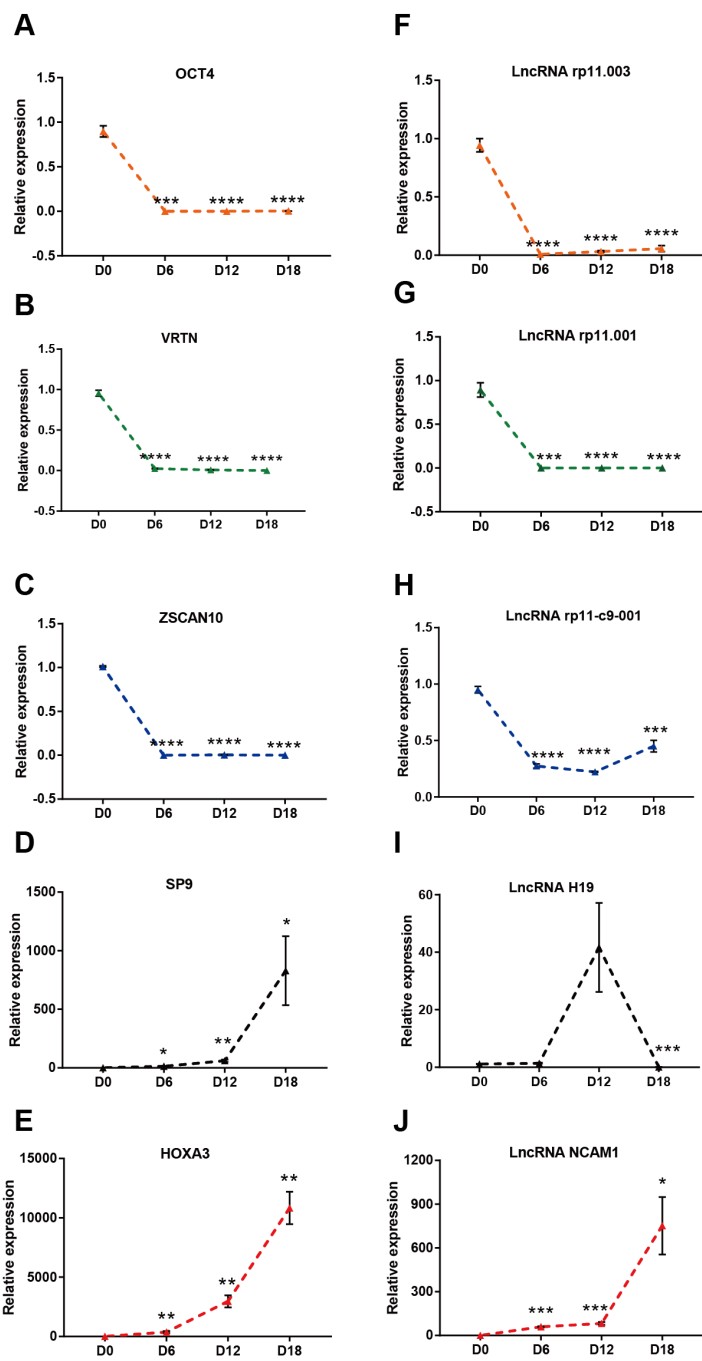

**Figure 4 Validation of DElncRNAs and mRNAs in MN differentiation.** (A–E) qRT-PCR analysis of the five DEmRNAs during MN differentiation. (F–J) qRT-PCR analysis of the five DElncRNAs up-regulated during MN differentiation * $P < 0.05$, ** $P < 0.01$, *** $P < 0.001$ and **** $P < 0.0001$ compared to D0. The number of each group was three to four.

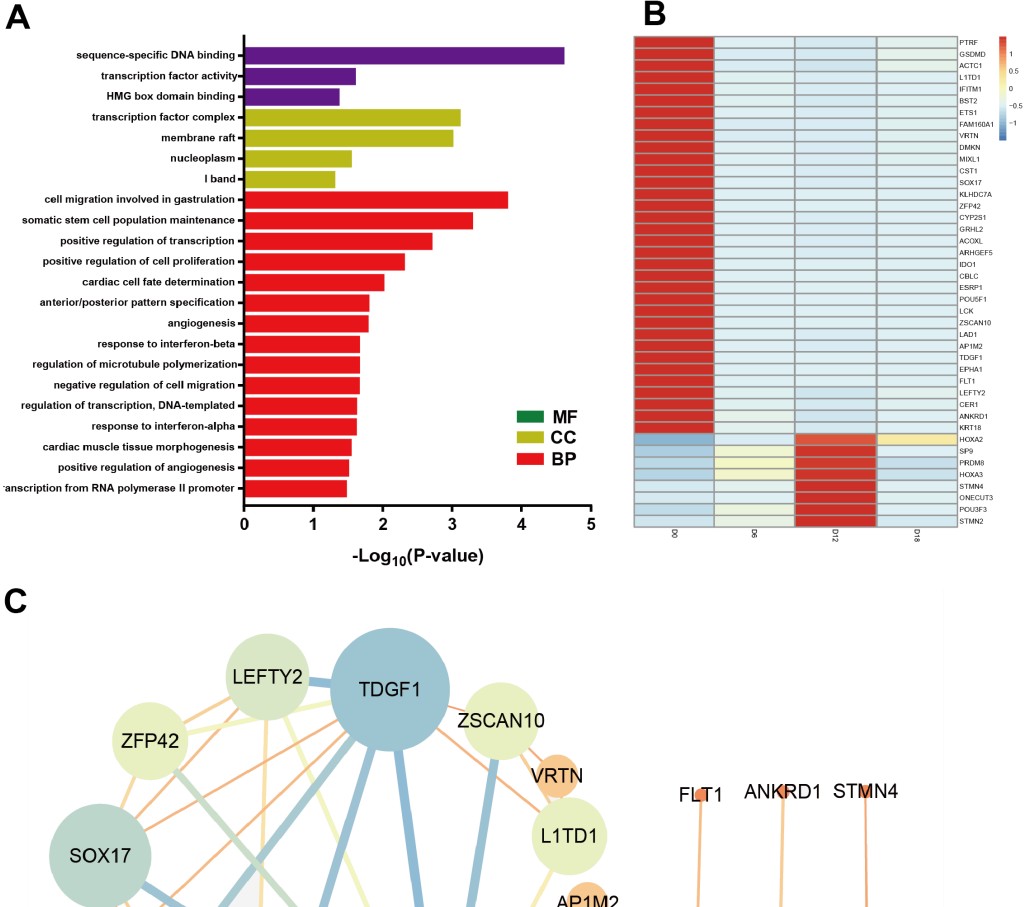

**Figure 5  Bioinformatic analysis of DEmRNAs the biological function.** (A) Go analysis of shared DEm-RNAs at D6, D12 and D18 compared with D0. (B) Heat map of shared DEmRNAs at D6, D12 and D18 compared with D0. (C) Protein interaction network analysis of shared DE mRNAs at D6, D12 and D18 compared with D0.

up-regulation in the transition from ESCs to MN stages, and especially showed a sharp up-regulation coincident with MNs specification. Interestingly, the expression of lncRNA NCAM1-AS showed same trend as HOXA3 and SP9. The NCAM1-AS is antisense lncRNA with two exons, produced from gene NCAM1, which is involved in cell adhesion, axonal outgrowth, synapse formation during development and differentiation, and highly expressed in the developing central and peripheral nervous systems (*Wobst et al., 2015*).
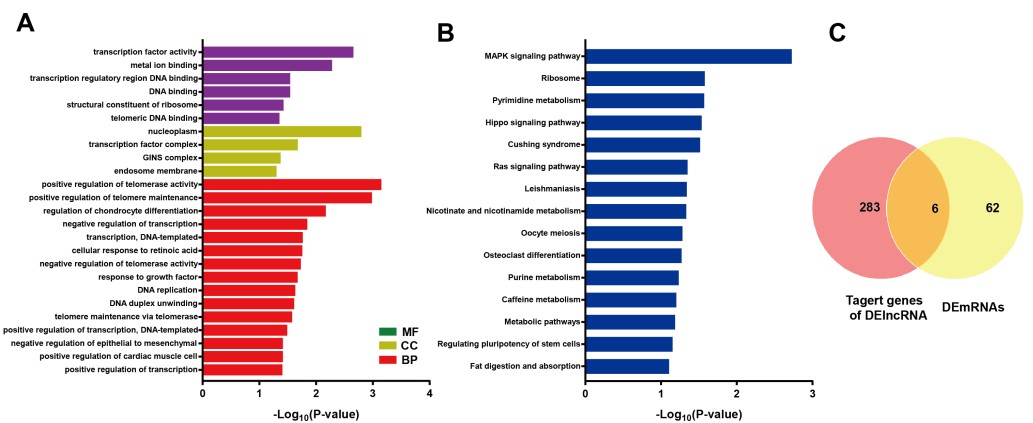

**Figure 6 Bioinformatic analysis of DElncRNAs the biological function.** (A) Go analysis of DElncRNAs at MN stage. (B) KEGG analysis of DElncRNAs at MN stage. (C) Venn diagram showed mRNAs tagerted by DElncRNA and DEmRNAs at the MN stage.

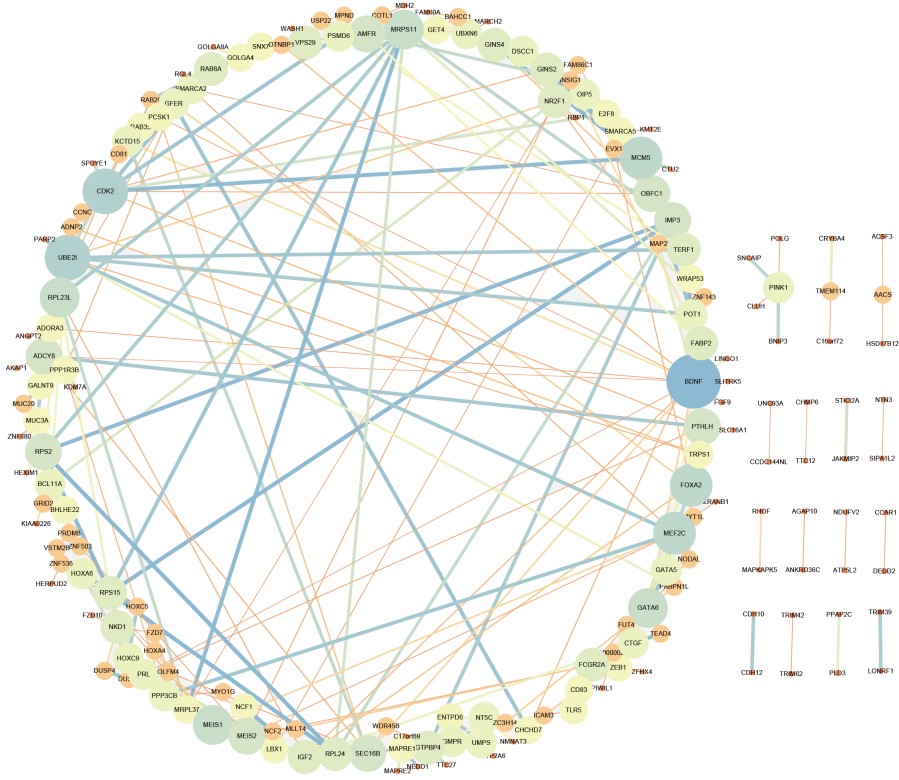

**Figure 7 Protein interaction network analysis of mRNAs targeted by DElncRNAs at the MN stage.**

Bioinformatics analysis of same putative target miRNAs of HOXA3 and lncRNA NCAM1-AS identified a direct binding site of has-miR-338-3p. Thus, lncRNA-miRNA-mRNA pathway was constructed including: lncRNA NCAM1-AS-miR-338-3p-HOXA3. Moreover,

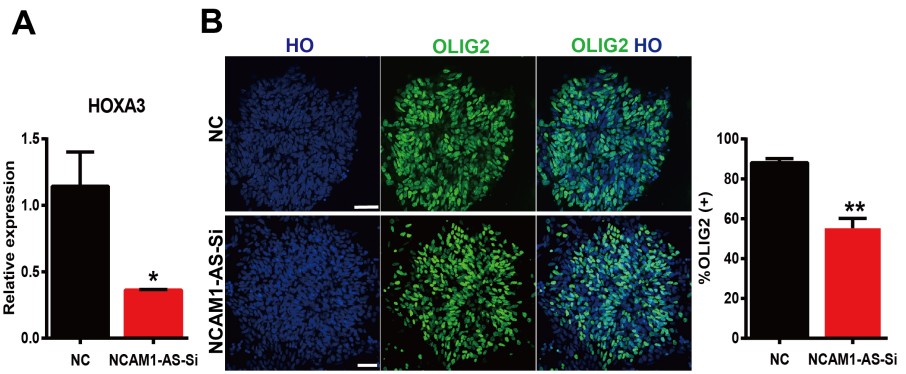

**Figure 8 Silencing of lncRNA NCAM1-AS inhibit MNP differentiation.** (A) The expression of HOXA3 was analyzed after NCAM1-AS knockdown. (B) Immunostaining of OLIG2 expression in the after NCAM1-AS knockdown at the MNP stage. Scare bar = 50 $\mu$m. * $P < 0.05$, ** $P < 0.01$.

the expression of HOXA3 was down-regulated at D12 in the NCAM1-AS Smart Silencer group compared to NC group (Fig. 8A). Furthermore, immunofluorescence analysis showed a decreased expression of OLIG2 in the NCAM1-AS knockdown group (Fig. 8B). Therefore, the results suggested that lncRNA NCAM1-AS can affect MNP differentiation.

## DISCUSSION

Motor neuron differentiation is precisely regulated and orchestrated by combinatorial expression of TFs during embryogenesis (*Alaynick, Jessell & Pfaff, 2011*). Accumulating evidence suggested that lncRNAs could interact with transcription factors to regulate cell differentiation (*Lopez-Pajares et al., 2015*; *Ng et al., 2013*; *Wang et al., 2014*). In this study, we profiled mRNAs and lncRNAs expression from our highly efficient ESC-derived MN differentiation protocol to study the development of MNs. Our analysis focused on the identification of transcription factors and lncRNAs that are strongly involved in the temporal development of MNs.

ESCs, derived from the inner cell mass of blastocyst stage embryos, can both self-renew and differentiate into other cell types (*Thomson et al., 1998*). The balance between self-renewal and differentiation is regulated by a complex interaction network of translation factors. Pluripotent genes such as Oct4, Nanog, Sox2, Klf4, and Myc (*Chambers & Tomlinson, 2009*), activated in ESCs, were inhibited during cell differentiation, whereas expression of differentiation marker genes increases gradually.

Notably, we found the well-known pluripotency-associated transcription factors POU5F1 (also known as OCT4) and TDGF1 which were hub downstream-regulated genes upon MN differentiation in our study. It has been reported that lincRNA linc-RoR, may function as a key ceRNA to link the network of miRNAs and core TFs OCT4, SPX2, and NANOG, thus regulating ESC maintenance and differentiation (*Wang et al., 2013*). As a previous study (*Zhang et al., 2006*), hub transcription factor ZSCAN10, verified by qRT-PCR, also down-regulated in our study and could regulate ESCs gene expression and

differentiation. OCT4 can directly regulate expression of ZSCAN10 and TDGF1 (*Wang et al., 2007*; *Babaie et al., 2007*).

Here, we also identified *cis* regulatory target genes HOXA6 and HOXC9 of lncRNA HOXA-AS3 and HOXC-AS2 at MN stage, respectively. Spinal MNs acquire specialized pool identities that guide their axons to target muscles in the limb, and the specificity of these precise connections (*Dasen et al., 2005*). MNs could express many HOX genes specifying MN pool identity and connectivity (*Dasen et al., 2005*; *Lacombe et al., 2013*). HOX6 paralog group genes (HOXA6, HOXC6, and HOXB6) contributed to diverse aspects of motor neuron subtype differentiation, and determined lateral motor column (LMC) fate at forelimb levels of the spinal cord (*Lacombe et al., 2013*). In addition, HOXC9 determined thoracic level MN population fates, including preganglionic column (PGC) and hypaxial motor column (HMC) neurons (*Jung et al., 2010*). The LncRNA HOXA-AS3 was found to inhibit osteogenic differentiation and promote adipogenic differentiation (*Wu et al., 2017*). The up-regulated lncRNAs HOXA-AS3 and HOXC-AS2 at human MN derived from ESCs was first identified in our study. The two lncRNAs might be involved in MN differentiation by *cis*-regulating HOXA6 and HOXC9, which needs further study.

Our sequencing results suggested a series of mRNAs and lncRNAs significantly changed during the transition from ESC to motor neurons. HOXA3, SP9 and lncRNA NCAM1-AS verified by PCR were observed dramatically up-regulated, especially at period of motor neuron. HOXA3 and HOXB3 are necessary for the specification of Pax6-and Olig2-dependent somatic MN progenitors (*Gaufo, Thomas & Capecchi, 2003*). In addition, HOX1 was reported to be involved in mediating both the role of RA-signaling in specification of hindbrain MNs (*Schubert et al., 2006*). In our sequencing data, HOXB3 was up-regulated at NEP and MNP stages but not altered at MN stage. The neuronal differentiation marker NCAM was involved in motor neurons functionally expanding synaptic territory (*Chipman, Schachner & Rafuse, 2014*).

Additionally, the potential function of lncRNA NCAM1-AS has been originally identified in human MN differentiation from ESC. The lncRNAs could function as miRNA sponges and might compete against other endogenous RNAs to regulate mRNA expression levels and maintain normal biological function. Bioinformatics analysis indicated lncRNA-miRNA-mRNA pathway: lncRNA NCAM1-AS-miR-338-3p-HOXA3. A recent study found that miR-338-3p targeted and inhibited HOXA3 in breast cancer (*Zhang & Ding, 2019*). Thus, the upregulated lncRNA NCAM1-AS might inhibit the expression of miRNA by acting as a miRNA sponge, and in turn, increasing the expressions of MN differentiation-associated mRNAs HOXA3. The dysregulated lncRNAs may regulate gene expression through many ways and play a critical role in the processes of neuronal differentiation.

## CONCLUSIONS

In conclusion, we utilized our highly efficient ESC-derived MNs differentiation protocol and next-generation sequencing to provide new insights into understanding the molecular mechanisms underlying MN differentiation and modulating lineage commitment of ESCs. The understanding of MN differentiation could ultimately offer the early diagnosis and

novel therapeutic tools of MN-related diseases. More importantly, additional experiments will be needed to further validate those lncRNAs functions.

## ACKNOWLEDGEMENTS

We acknowledge Professor Su-Chun Zhang of University of Wisconsin–Madison for his assistance during the development of this work.

### Funding

This work was funded by the National Key Research and Development Project of China (2016YFA0102500), the National Natural Science Foundation of China (81801066), and the China Postdoctoral Science Foundation (2018M640706). The funders had no role in study design, data collection and analysis, decision to publish, or preparation of the manuscript.

### Grant Disclosures

The following grant information was disclosed by the authors:
National Key Research and Development Project of China: 2016YFA0102500.
National Natural Science Foundation of China: 81801066.
China Postdoctoral Science Foundation: 2018M640706.

### Competing Interests

The authors declare there are no competing interests.

### Author Contributions

- Xue-Jiao Sun conceived and designed the experiments, performed the experiments, analyzed the data, prepared figures and/or tables, authored or reviewed drafts of the paper, and approved the final draft.
- Ming-Xing Li performed the experiments, analyzed the data, prepared figures and/or tables, authored or reviewed drafts of the paper, and approved the final draft.
- Chen-Zi Gong performed the experiments, authored or reviewed drafts of the paper, and approved the final draft.
- Jing Chen performed the experiments, analyzed the data, prepared figures and/or tables, and approved the final draft.
- Mohammad Nasb and Muhammad Rehan analyzed the data, authored or reviewed drafts of the paper, and approved the final draft.
- Sayed Zulfiqar Ali Shah analyzed the data, prepared figures and/or tables, authored or reviewed drafts of the paper, and approved the final draft.
- Ya-Jie Li performed the experiments, analyzed the data, authored or reviewed drafts of the paper, and approved the final draft.
- Hong Chen conceived and designed the experiments, authored or reviewed drafts of the paper, and approved the final draft.

## Data Availability

Raw RNA sequencing data are available in the public Gene Expression Omnibus (GEO) database: GSE151744.

lncRNA sequence is available at Figshare: Sun, Xuejiao (2020): Temporal expression profiles of lncRNA and mRNA in human embryonic stem cell-derived motor neurons during differentiation. figshare. Dataset. https://doi.org/10.6084/m9.figshare.9948179.v1.

## Supplemental Information

Supplemental information for this article can be found online at http://dx.doi.org/10.7717/peerj.10075#supplemental-information.

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
