# Peer review of "Temporal expression profiles of lncRNA and mRNA in human embryonic stem cell-derived motor neurons during differentiation"

_PeerJ, doi:10.7717/peerj.10075_

## Round 0.1 · original submission · Major Revisions

It looks that there are major issues in the manuscript. Please refer to the referees' reports for details.

·

Basic reporting

English should be carefully checked by native English speakers. This manuscript is very descriptive and no confirmation of the impact of DE-lnc RNA and mRNA in the MN differentiation.

Experimental design

In this paper, research question was well defined, and it is meaningful in the research field. Methods are well described in this study with adequate information and references. However, I strongly recommend validating the role of newly identified key gene(s)/lncRNAs using siRNA-mediated knockdown whether it would change MN differentiation capacity (by checking proper markers gene expression or dendritic arborization).

Validity of the findings

This study provides reproducible analysis and its impact is important for the field of neuroscience research. The topic is interesting however further confirmation is required to validate their hypothesis regarding the roles of lncRNA in the MN differentiation.

Additional comments

The author bioinformatically analyzed the transcript sequencing data of motor neuron (MN) differentiation. This paper is well written, and its analysis is clear. However, functional studies (as suggested) should be performed to validate the bioinformatic analysis. For example, is there any data previously published data to support the gene profiling of the MN differentiation (especially lnc RNA expression pattern)? Are there any studies using in vivo models regarding the identified pathway in this study? If so, please refer to the papers in this study. If not, please knockout or knockdown the identified RNAs to be involved in the MN fifferntiation and check the effect of them on MN differentiation (Just to prove this study is valid). As they aslo use a cell line for ES cells, I believe it should be required to confirm these things (please test some gene knockdown or knockout experiments).

Reviewer 2 ·

Basic reporting

1. There are many problems with sentence structure, verb tense, and clause construction in the paper, which needs to be proof-read carefully and checked. The English of your manuscript must be improved before resubmission. For example, at the first line in abstract, “have” should be deleted; at the 35th line, “predict” should be revised as “predicted”; at the 37th line, “construct” should be revised as “constructed”; at the 40th line, “the” is advised to be changed to “a”; at the 72th line, “mRANs” should be revised as “mRNAs” etc.
2. At the 129th line, Figure 2 should be changed to Figure 1B.
3. Please check the 152th line and 155th line to confirm Figure 4A and Figure 4B. Figure 4A and 4B should be replaced with each other.
4. Please add the Kyoto Encyclopedia of Genes and Genomes (KEGG) pathway enrichment analysis and Heat map for the differentially expressed mRNA.
5. Function enrichment analysis is advised to do for the differentially expressed mRNA and lncRNA respectively.
6. Please show the clearer image of Figure 6, especially for Figure 6C.
7. It is advised to be confirmed in cells whether has-miR-338-3p inhibits the processes of motor neurons differentiation through targeting HOXA3, which is ameliorated by lncRNA NCAM1-AS.

Experimental design

Reasonable

Validity of the findings

1. Statistical tests should be stated in the figure legends. The number of determinations analyzed in each group of each graph should be stated in the Figure legends.
2. In Figure 4, please show the statistical meaning of “*” in Figure legend.

Additional comments

The authors have identified lncRNAs and mRNAs during motor neurons differentiation of human embryonic stem cells. From 283 target genes of differentially expressed lncRNAs, six mRNAs were found to be differentially expressed, and lncRNAs (NCAM1-AS) and mRNAs (HOXA3) were further confirmed for constructing the lncRNA NCAM1-AS-miRNA-HOXA3 network using bioinformatics analyses. The experiment design and data are sound, but the manuscript is far from qualified to publish in Peer J now. There are some suggestions above that the authors should address to help improve the quality of the manuscript.

·

Basic reporting

no comment

Experimental design

no comment

Validity of the findings

no comment

Additional comments

This paper mainly shows temporal expression profiles of lncRNA and mRNA related to MN differentiation, which seems important in uncovering the mechanism of MN differentiation. However, based on the current data of this paper, I do believe is not good enough for publishing, and I don’t advise the editor to accept this paper considering the follow major related problems:
1. We think the method by which the author discovered the different expression genes is somewhat improper. The process that hESCs differentiate into spinal cord MNs is a kind of dynamic regulation. Though the author compare D6, D12, D18 to D0, respectively, the comparion between D12 and D6, D18 and D12 is necessary as well. Because there may exist some genes up-regulated at the beginning, then down-regulated later.
2. In Figure 4A, the author said “However, lncRNA H9 increased significantly at D12, but decreased at D18. Our results agreed with the data of RNA-sequencing generally”. The author only found the up-regulated or down-regulated lncRNAs, why did they conclude that the dynamic expressed H9 agreed with the data of RNA-sequencing?
3. The author identified the key factors that regulated MN differentiation by GO analysis. However, the author only listed the Go terms, failed to discover the underlying meaning of GO analysis.
4. In Figure 5B, the author only drew the PPI network, but failed to explain the result of this analysis. Though we saw the relationship between two or more genes, what’s the importance of them during MN differentiation?
5. In Figure 6C, the author said “We also performed the topological analysis of PPI network on these target genes”. Where is the conclusion?
6. We can’t find the figure which shows “Construction of lncRNA -miRNA-mRNA interaction network”.
7. We have to say the writing of this paper is a little careless. Particularly, in section Results, the author only listed the results mostly, but failed to draw conclusions.
8. There is a mistake in Table 1, HOXA3 and SP9 ‘Reverse’part.

---

## Round 0.2 · Minor Revisions

Please address the remaining concerns raised by the reviewer.

·

Basic reporting

English is improved, however I would like to see one example of the effect of lncRNA/miRNA (the ones they identified in this study) in the process of neuronal differentiation. I think it does not take that long procedure to validate the effect of one lncRNA/miRNA.

Experimental design

The same as written in basic report. The authors should validate the role of one of the lncRNA in neuronal differentiation.

Validity of the findings

N/A(mentioned in previous round of review)

Additional comments

I still think it would be nice to have one good example of what happens if they deplete one/ a few lncRNA and perform a simple experiment (eg. compare using proper stem and neuronal marker gene expressions). The best comparison is D0 vs D18 but, D0 vs D6 is an alternative if they cannot wait for 18days to get full differentiation status).

---

## Round 0.3 · Major Revisions

The reviewer has asked the authors to show how does lncRNA/miRNA (the ones they identified in this study) control the process of neuronal differentiation. As an editor, I have easily speculated that the authors would present some visual image of motor neurons whose differentiation is controlled by the lncRNA/miRNA. The authors instead provided expression profiles, which is less informative than I have expected. I would encourage the authors to show how the motor neuron differentiation could be altered by suppressing a relevant ncRNA.

---

## Round 0.4 · accepted · Accept

The manuscript is accepted, but the authors need to correct some typos identified by Reviewer 2.

·

Basic reporting

all the criteria is fulfilled.

Experimental design

all the criteria is fulfilled.

Validity of the findings

all the criteria is fulfilled.

Additional comments

The authors have added new data regarding the effect of lncRNA NCAM1-AS on MNP differentiation. Therefore, now the quality of the study is at the level of the publication in Peer J.

Reviewer 2 ·

Basic reporting

The authors basically have addressed the point that raised. But there are still some writing mistakes that authors should carefully correct. For example, at the 76th line, “form” should be revised as “from”; at the 126th line, “was” should be revised as “were” etc.

Experimental design

Reasonable

Validity of the findings

No comment

Additional comments

1. The authors should respond point-to-point.
2. The authors basically have addressed the point that raised. But there are still some writing mistakes that authors should carefully correct. For example, at the 76th line, “form” should be revised as “from”; at the 126th line, “was” should be revised as “were” etc. After the authors have corrected the grammar and writing mistakes, the article can be considered to publish in the Peer J.